# Molecular Characterisation of *Cryptosporidium* spp. in Mozambican Children Younger than 5 Years Enrolled in a Matched Case-Control Study on the Aetiology of Diarrhoeal Disease

**DOI:** 10.3390/pathogens10040452

**Published:** 2021-04-09

**Authors:** Augusto Messa, Pamela C. Köster, Marcelino Garrine, Tacilta Nhampossa, Sérgio Massora, Anélsio Cossa, Quique Bassat, Karen Kotloff, Myron M. Levine, Pedro L. Alonso, David Carmena, Inácio Mandomando

**Affiliations:** 1Centro de Investigação em Saúde de Manhiça, Maputo 1929, Mozambique; augusto.junior@manhica.net (A.M.J.); marcelino.garrine@manhica.net (M.G.); tacilta.nhampossa@manhica.net (T.N.); sergio.massora@manhica.net (S.M.); anelsio.cossa@manhica.net (A.C.); quique.bassat@isglobal.org (Q.B.); alonsop@who.int (P.L.A.); 2Parasitology Reference and Research Laboratory, National Centre for Microbiology, Health Institute Carlos III, Majadahonda, 28220 Madrid, Spain; pamkoste@ucm.es; 3Global Health and Tropical Medicine, Instituto de Higiene e Medicina Tropical, Universidade Nova de Lisboa, 1349-008 Lisbon, Portugal; 4Instituto Nacional de Saúde, Ministério da Saúde, Marracuene, Maputo 1120, Mozambique; 5ISGlobal, Hospital Clínic—Universitat de Barcelona, 08036 Barcelona, Spain; 6ICREA, Pg. Lluís Companys 23, 08010 Barcelona, Spain; 7Pediatric Infectious Diseases Unit, Pediatrics Department, Hospital Sant Joan de Déu, University of Barcelona, 08950 Barcelona, Spain; 8Consorcio de Investigación Biomédica en Red de Epidemiología y Salud Pública (CIBERESP), 28029 Madrid, Spain; 9Center for Vaccine Development, University of Maryland School of Medicine, Baltimore, MD 21201-1509, USA; kkotloff@som.umaryland.edu (K.K.); mlevine@som.umaryland.edu (M.M.L.); 10Global Malaria Program, World Health Organization, 1211 Geneva, Switzerland

**Keywords:** *Cryptosporidium*, *gp60*, *ssu* rRNA, genotyping, children, diarrhoea, prevalence, molecular epidemiology, Mozambique, GEMS

## Abstract

*Cryptosporidium* is a leading cause of childhood diarrhoea and associated physical and cognitive impairment in low-resource settings. *Cryptosporidium*-positive faecal samples (*n* = 190) from children aged ≤ 5 years enrolled in the Global Enteric Multicenter Study (GEMS) in Mozambique detected by ELISA (11.5%, 430/3754) were successfully PCR-amplified and sequenced at the *gp60* or *ssu* rRNA loci for species determination and genotyping. Three *Cryptosporidium* species including *C. hominis* (72.6%, 138/190), *C. parvum* (22.6%, 43/190), and *C. meleagridis* (4.2%, 8/190) were detected. Children ≤ 23 months were more exposed to *Cryptosporidium* spp. infections than older children. Both *C. hominis* and *C. parvum* were more prevalent among children with diarrhoeal disease compared to those children without it (47.6% vs. 33.3%, *p* = 0.007 and 23.7% vs. 11.8%, *p* = 0.014, respectively). A high intra-species genetic variability was observed within *C. hominis* (subtype families Ia, Ib, Id, Ie, and If) and *C. parvum* (subtype families IIb, IIc, IIe, and IIi) but not within *C. meleagridis* (subtype family IIIb). No association between *Cryptosporidium* species/genotypes and child’s age was demonstrated. The predominance of *C. hominis* and *C. parvum* IIc suggests that most of the *Cryptosporidium* infections were anthroponotically transmitted, although zoonotic transmission events also occurred at an unknown rate. The role of livestock, poultry, and other domestic animal species as sources of environmental contamination and human cryptosporidiosis should be investigated in further molecular epidemiological studies in Mozambique.

## 1. Introduction

Diarrhoeal diseases remain the second leading cause of mortality after pneumonia in children under 5 years worldwide, accounting for approximately 9% of the 5.8 million deaths associated to this condition reported in 2015 [1,2]. Most of these fatalities disproportionately occur in poor-resource settings where suboptimal hygiene conditions and sanitation prevail [3]. A recent update on diarrhoeal burden from the Global Enteric Multicenter Study (GEMS and GEMS1A) demonstrated that Rotavirus, *Cryptosporidium*, enterotoxigenic *Escherichia coli* producing heat stable toxin (ST_ETEC), and *Shigella* were the main pathogens associated with moderate-to-severe diarrhoea (MSD) and less severe diarrhoea (LSD) in African and Asian children [4,5]. Cryptosporidiosis in children under 5 years presents with watery diarrhoea, abdominal pain, nausea, and vomiting, being often associated to growth faltering and cognitive development impairment [6,7]. In severe cases, the disease can lead to life-threatening sequelae among malnourished and immunocompromised children [8]. *Cryptosporidium* spp. is also a major contributor to the burden of diarrhoeal disease in HIV-positive patients [9].

As other diarrhoea-causing pathogens, *Cryptosporidium* spp. are transmitted through the faecal–oral route. Humans acquire the infection through direct contact with infected hosts (person-to-person or zoonotic transmission) or by ingestion of contaminated food or water (foodborne and waterborne transmission), but the relative importance of these routes is still unclear [10]. At least 40 known *Cryptosporidium* species are currently recognised, and among these, more than 20 species and genotypes have been reported to cause human infections. *Cryptosporidium hominis* and *C. parvum* cause more than 90% of the human cases documented globally [11,12,13].

Molecular tools for the differentiation of *Cryptosporidium* species and genotypes are currently available, mostly using PCR followed by either restriction length fragment polymorphisms (RFLP) analysis or Sanger sequencing of the small subunit ribosomal ribonucleic acid (*ssu* rRNA) and the 60 kDa glycoprotein (*gp60*) genes of the parasite [10,13]. The *ssu* rRNA gene is largely used for the differential diagnosis of *Cryptosporidium* species due to its multicopy nature and associated high sensitivity. Subtype identification is primarily achieved through DNA sequence analysis of the highly polymorphic *gp60* gene. Subtype assignment is based on the number of TCA, TCG, and TCT repeats in addition to other repetitive sequences, such as the ACATCA, within the *gp60* tandem repeat motif region. Subtype families are named as Ia, Ib, Ic, Id, Ie, If, etc. for *C. hominis* and IIa, IIb, IIc, IId, etc. for *C. parvum*, with further species families named in ascending order [12,14].

In Africa, the epidemiology and genetic diversity of *Cryptosporidium* spp. remains relatively unknown. However, as noted by a recent literature review, at least 13 species and genotypes have been identified in humans, with *C. hominis* followed by *C. parvum* once again dominating the epidemiological landscape [11]. Subtyping studies support the dominance of anthroponotic over zoonotic transmission in African countries, regardless of the close contact with farm and domestic animals. Another interesting observation in Africa is the high level of subtype diversity, where at least six subtype families for *C. hominis* (Ia, Ib, Id, Ie, If, and Ih) have been described. For *C. parvum* the predominant subtypes identified in humans belong to the IIc family, in addition to IIa, IIb, IId, IIg, IIi, IIh, IIm and the rarer anthroponotically transmitted IIe subtype family [11].

In Mozambique, diarrhoea is ranked as the third cause of death in children under 14 years from the capital city Maputo [15] and fourth in children under 5 years from the Manhiça district (Maputo province) [16], being responsible for 20% of hospital paediatric admissions in this district [17]. The GEMS data support that prevention strategies targeting Rotavirus, *Cryptosporidium*, ST_ETEC and *Shigella* could contribute to reduce diarrhoeal cases by approximately 50% in infants, and hence diarrhoeal-associated mortality [18]. However, the genetic diversity of *Cryptosporidium* spp. in GEMS was not investigated. Two previous hospital-based studies carried out in the southern part of the country have identified IaA23R3, IIcA5G3, and IIeA12G1 subtypes among nine isolates from patients with diarrhoea in the capital city Maputo [19], and IbA10G2 and IdA22 subtypes among eight isolates in patients with HIV and tuberculosis in the Chokwe district of Gaza province [20]. However, no extensive molecular epidemiological studies have been conducted to evaluate the genetic diversity within *Cryptosporidium* spp. Herein, we aimed to analyse the diversity and frequency of *Cryptosporidium* species and subtypes detected in stools from children younger than 5 years from the Manhiça district, Mozambique, enrolled in the context of GEMS between 2007 and 2012.

## 2. Results

### 2.1. Initial Screening for the Detection of Cryptosporidium spp. by ELISA Immunoassay

During the 5-year study period (December 2007–November 2012) a total of 3754 stool samples were collected. The ELISA positivity rate for *Cryptosporidium* spp. was estimated at 11.5% (430/3754). The prevalence was significantly (*p* < 0.001) higher among diarrhoea cases (MSD and LSD cases) (16.5%, 222/1346) compared to children without diarrhoea (non-cases; 8.5%, 208/2408). Most (91.2%, 392/430) of the *Cryptosporidium*-positive samples by ELISA were available for molecular analyses (Figure 1). Unavailable samples were the result of the depletion of starting material as consequence of testing and analyses in previous studies [4,5,18].

The distribution of the ELISA-positive *Cryptosporidium* infections in cases and non-cases according to sex, age group, and clinical condition is summarised in Table 1. Approximately one in two (53.6%, 210/392) children with cryptosporidiosis were aged 0–11 months. The male/female ratio was 1.8. Children with MSD and their matched controls were significantly more exposed to *Cryptosporidium* than their counterparts with LSD and corresponding controls (*p* < 0.001). HIV+ patients with diarrhoea were more likely to be infected with *Cryptosporidium* spp. than HIV+ patients without diarrhoea (χ^2^ = 9.8758, *p* = 0.001675). Being undernourished and having diarrhoea were also significantly associated with cryptosporidiosis (χ^2^ = 19.769, *p* ≤ 0.00001). Regarding coinfections with other intestinal pathogens, *Cryptosporidium* infection was more likely in children with diarrhoea and rotavirus infection (*p* ≤ 0.011). In contrast, coinfections by *Cryptosporidium* spp. and *G. duodenalis* were more frequent in asymptomatic (non-cases) children (*p* < 0.001). The full dataset showing the epidemiological, clinical, diagnostic, and genotyping data used in the analyses conducted in the present survey is presented in Appendix A.

### 2.2. Confirmation of Cryptosporidium spp. by Nested PCR Methods

Out of the 392 samples that tested positive by ELISA, 37.2% (146/392) were successfully sub-genotyped at the *gp60* locus. The remaining 250 isolates with a negative result by *gp60*-PCR were subsequently re-assessed at the *ssu* rRNA marker, allowing the confirmation of *Cryptosporidium* DNA in 44 additional samples. Overall, the presence of the parasite was confirmed by *gp60*-PCR and/or *ssu*-PCR in 48.5% (190/392) of the analysed samples (Table 2). Sequence alignment analyses including appropriate reference sequences allowed the identification of three *Cryptosporidium* species including *C. hominis* (72.6%, 138/190), *C. parvum* (22.6%, 43/190), and *C. meleagridis* (4.2%, 8/190). An additional isolate (0.5%, 1/190) was only identified at the genus level due to poor sequence quality (Table 2). Both *C. hominis* and *C. parvum* were more prevalent among diarrhoeal children (cases) compared to non-diarrhoeal (non-cases) children (47.6% vs. 33.3%, *p* = 0.007 and 23.7% vs. 11.8%, *p* = 0.014, respectively). Infections by *Cryptosporidium* spp. were most common in children younger than 24 months, with *C. hominis* being the *Cryptosporidium* species more prevalent in all age groups investigated (Table 2). Cases of cryptosporidiosis by *C. hominis* and *C. parvum* were consistently detected along the whole study period, peaking during November 2011 and March 2012, particularly in children with LSD (Appendix A).

### 2.3. Genetic Variation within C. hominis and C. parvum

Sequence analysis of the 117 isolates characterised as *C. hominis* at the *gp60* locus revealed the presence of five subtype families including Ia (35.0%, 41/117), Ib (20.5%, 24/117), Id (1.7%, 2/117), Ie (34.2%, 40/117), and If (8.6%, 10/117). The most prevalent subtypes found were IaA23R1 within family Ia, IbA13G2 within family Ib, and IdA20 within family Id. All isolates belonging to families Ie and If were identified as IeA11G3T3 and IfA12G1, respectively. Two genetic variants within IaA24R3 and IbA13G2 corresponded to novel subtypes (Table 3). Sequence analyses of the 29 isolates characterised as *C. parvum* at the same locus revealed the presence of four subtype families including IIb (3.4%, 1/29), IIc (86.2%, 25/29), IIe (6.9%, 2/29), and IIi (3.4%, 1/29). IIbA11 within family IIb, IIcA5G3 within family IIc, IIeA11G1 within family IIe, and IIiA6-like within family IIi were the only subtypes found. Novel genetic variants were found within IIbA11, IIcA5G3, and IIiA6-like subtypes. All four isolates assigned to *C. meleagridis* belonged to subtype IIIbA23G1R1 within family IIIb (Table 3).

Out of the 21 sequences characterised as *C. hominis* at the *ssu* rRNA locus, 81% (17/21) showed 100% identity with reference sequence AF108865. The remaining four sequences differed from AF108865 by 1–3 single nucleotide polymorphisms (SNPs) including a deletion mutation (Table 3). All the 14 sequences assigned to *C. parvum* corresponded to different genetic variations of the “bovine genotype” of this *Cryptosporidium* species, which is characterised by the presence of a four-base deletion TAAT at positions 686 to 689 of reference sequence AF112571. Indeed, some authors have proposed this genetic variant as an independent species named *C. pestis* [22]. *Cryptosporidium hominis* and *C. parvum* sequences differing from reference sequences at the *ssu* rRNA locus included ambiguous (C/T, A/G) positions in the form of double peaks, transition (A↔G, C↔T) and transversion (T↔G, A↔T) mutations, and single- to multiple-base deletions.

Finally, all eight sequences identified as *C. meleagridis* at the *ssu* rRNA locus showed 100% identity with reference sequence AF112574. Four of these eight isolates were successfully amplified at the *gp60* locus using a specific PCR protocol for this *Cryptosporidium* species (see Section 4.3.2.). Sanger sequencing analyses allowed the identification of subtype IIIbA23G1R1 in all four sequences, which were identical to reference sequence MK331716.

*Cryptosporidium hominis* was the most prevalent species in children with MSD or LSD (79.6%, 70/88), followed by *C. parvum* (19.3%, 17/88), and *C. meleagridis* (1.1%, 1/88) (Table 4). Within *C. hominis*, nearly three out of every four diarrhoea-associated infections were caused by subtype families Ie (30.7%, 27/88) and Ia (27.3%, 24/88). Subtype family Ie was more frequent in children with MSD (43.1%, 25/58), and subtype family Ia in children with LSD (46.7%, 14/30). No obvious differences in subtype distribution were observed among the age groups considered. Near half of the cryptosporidiosis cases identified in HIV+ patients were caused by the subtype family Ia (55.6%, 5/9) (Table 4).

In matched controls without diarrhoea (non-cases), *C. hominis* was also the predominant species found (75.0%, 45/60), followed by *C. parvum* (20.0%, 12/60) and *C. meleagridis* (5.0%, 3/60) (Appendix A). Within *C. hominis*, half of the infections were attributed to subtype families Ia (26.7%, 16/60) and Ib (21.7%, 13/60). No obvious differences in subtype distribution were observed among the age groups considered. No *Cryptosporidium* subtype families could be determined in HIV+ patients without diarrhoea (Appendix A).

*Cryptosporidium hominis* subtype families Ib and Ie were more frequently found during study years 1 to 4, whereas *C. hominis* subtype family Ia and *C. parvum* subtype family IIc were observed only in study year 5, suggesting variable seasonal patterns in the frequency of *Cryptosporidium* subtypes (Appendix A).

The genetic relationships among *gp60* gene sequences generated in the present study, as inferred by a neighbor-joining analysis, are shown in Figure 2. All *Cryptosporidium* sequences clustered together (monophyletic groups) with different well-supported clades (≥93% of bootstrap) corresponding to appropriate reference sequences for *Cryptosporidium* subtype families.

## 3. Discussion

This is the most comprehensive molecular epidemiological study conducted in Mozambique to date investigating the genetic diversity of the diarrhoea-causing enteric protozoan parasite *Cryptosporidium* spp. The analysis took advantage of the large sample repository generated by the GEMS in children younger than five years of age with and without diarrhoea in Maputo province [4,5]. Consequently, a total of 392 stool samples with a positive result by ELISA were available for molecular investigations, of which 190 were successfully genotyped at the *gp60* or *ssu* rRNA loci.

A preliminary assessment of the *Cryptosporidium*-positive samples by ELISA corroborated results obtained in previous epidemiological studies carried out in sub-Saharan African countries [11]. For instance, cryptosporidiosis was confirmed as a serious public health concern in children younger than 2 years old, particularly if immunocompromised by other infections (e.g., HIV/AIDS) or malnourished. Young children are more susceptible to intestinal parasites and other infectious pathogens due to their low level of immunity [23]. The level of exposure and risk of infections increase in poor settings with limited access to safe drinking water, sanitation, and hygiene [24,25]. A significant association between *Cryptosporidium* infection and malnutrition (stunting, wasting, underweight) has been documented in children from several African countries, including Kenya [26], Mozambique [27], Tanzania [28], and Uganda [29], among others. Similarly, *Cryptosporidium* infection was more frequently found in HIV-positive than in HIV-negative children and patients in Mozambique [20,30] and other African endemic regions [28,31,32].

Our molecular analyses revealed the presence of three *Cryptosporidium* species (*C. hominis*, *C. parvum*, and *C. meleagridis*) in the studied paediatric population. Mostly anthroponotic *C. hominis* and zoonotic *C. parvum* were previously known to be circulating in Mozambique [19,20,33], but this is the first report of *C. meleagridis* in the country. An additional two species, *C. felis* and *C. viatorum*, have also been recently described in adult patients with diarrhoea in the Maputo province and in asymptomatic children in the Zambézia province [19,34]. *Cryptosporidium meleagridis* and *C. felis* are adapted to infect birds and domestic cats, respectively, as primary host species, but they are also responsible for a significant number of human infections globally, particularly the former [10,12]. These data seem to indicate that direct contact with cats, poultry, and other avian species (or their faecal material) may be a risk factor for human cryptosporidiosis in Mozambique. Following the same line of reasoning, the fact that all the *C. parvum* isolates characterised at the *ssu* rRNA gene belonged to the “bovine genotype” of *C. parvum* supported the notion that an unknown number of human cases of cryptosporidiosis are indeed of zoonotic nature. This is without precluding that some of the infections caused by this genetic variant of *C. parvum* may be also transmitted through person-to-person contact. The extent of the exact contribution of each potential transmission pathway (zoonotic, anthropic, direct contact, or indirect through ingestion of contaminated water or food) remains to be elucidated. Finally, *C. viatorum* was initially thought to be a human-adapted species [35], but recent epidemiological surveys conducted in Australia and China have demonstrated that this *Cryptosporidium* species can successfully infect rodents and therefore may have zoonotic potential [36,37]. Overall, these data agree with those reported in the African continent, where *C. hominis* was the most prevalent (2–100%) *Cryptosporidium* species in humans, followed by *C. parvum* (3–100%) and *C. meleagridis* (up to 75%), the latter species found mainly in immunocompromised individuals [11].

Subtyping analyses identified Ia and Ie as the most prevalent subtype families within *C. hominis*, being responsible for nearly 70% of the infections attributable to this *Cryptosporidium* species in both diarrhoeal and non-diarrhoeal children. Similar results have been reported in Kenyan young children with and without HIV infection [38], mostly HIV-positive patients in Nigeria [39], children younger than 10 years in São Tomé and Príncipe [40], and children younger than five years in South Africa [41]. As already described in most previous epidemiological studies conducted in the continent, subtype family Ib was also underrepresented in the paediatric population surveyed here. Indeed, Ib has been shown as the predominant subtype family only in Nigerian children [42]. Subtype families If and Id are typically documented at low frequencies in African human populations. Subtype family If has been identified in Kenyan patients with and without HIV/AIDS [38,43], children of young age in South Africa [41], and individuals from rural areas in Tanzania [44]. Finally, members of the subtype family Id have been found circulating in HIV-positive patients in Ethiopia, Equatorial Guinea, and Mozambique [20,45,46], in children with diarrhoea in Ghana and Madagascar [47,48], in Kenyan children with and without HIV infection [38], and in paediatric populations from Nigeria and South Africa [41,42].

Mainly transmitted anthroponotically, IIc was the predominant (86%) *C. parvum* subtype family circulating in the children investigated here. This is in agreement with previous results obtained in diverse human populations from other sub-Saharan countries including Kenya [38], Madagascar [48], Nigeria [42], South Africa [41], and Uganda [49]. In contrast, subtype families IIb, IIe, and IIi were only sporadically detected, and subtype family IIa was absent. It should be noted that IIa was the most prevalent *C. parvum* subtype family circulating in HIV-positive and diarrhoeal disease patients in Ethiopia and Kenya [43,45,50] and also in Tunisian young children [51]. Taken together, these geographically segregated patterns of *C. parvum* genetic variants may be indicative of differences in sources of infections and transmission pathways.

Very limited information is currently available on the intra-species molecular diversity of *C. meleagridis* in African isolates of human origin. In the present study, all the *C. meleagridis* isolates identified belonged to the subtype family IIIb, and no genetic heterogeneity was observed among their sequences. This very same subtype family has also been reported in an urban population in Tunisia [52], whereas IIId has been described in diarrhoeic paediatric patients in South Africa [41]. Interestingly, a wide range of *C. meleagridis* subtype families including IIIb (but not IIId) has been recently identified in river water and its sediment in South Africa [53]. This finding has important public health implications, as it demonstrated that the consumption of contaminated, non-treated surface waters might lead to waterborne cryptosporidiosis by *C. meleagridis*.

The main strength of this study is the large number of *Cryptosporidium*-positive samples of human origin molecularly characterised by Sanger sequencing. However, certain methodological and study design issues may have hampered its accuracy. For instance, the fact that only half (48.6%, 190/392) of the ELISA-positive samples were amplified at the *gp60* or *ssu* rRNA loci may have biased the actual proportion of *Cryptosporidium* species and genotypes reported here. This may be due to the suboptimal preservation of parasitic DNA through time (stool samples were collected during the period 2007–2012), or to potential false-positive results in the ELISA immunoassay, or to amplification failure associated to suboptimal removal of PCR inhibitors (e.g., proteases, DNases, polysaccharides, bile salts). We cannot completely rule out the possibility that the ELISA immunoassay initially used for screening purposes yielded an unknown number of false-negative results, particularly for *Cryptosporidium* species less frequently found in humans (e.g., *C. felis*, *C. viatorum*, *C. ubiquitum*, among others). Additionally, no attempts were carried out to analyse in depth the potential associations between *Cryptosporidium* species/genotypes and the sociodemographic, epidemiological, and clinical features of the participating children, as this task will be specifically tackled in an independent study.

Overall, the high level of genetic diversity observed within *Cryptosporidium* isolates reveals an epidemiological scenario where infection and re-infection events seem common and environmental contamination high. In this regard, a recent risk association study conducted in the province of Zambézia revealed that drinking untreated water and having regular contact with domestic animals were major risks for acquiring protist infections including cryptosporidiosis [25]. Additionally, a recent quantitative microbial risk assessment analysis has estimated that the consumption of unsafe water causes 2 million cryptosporidiosis cases and 1.6 × 10^5^ disability-adjusted life years in Mozambique annually [54]. These results highlight the relevance of improving access to safe drinking water and sanitary conditions to minimise the risk of environmental contamination and the waterborne and foodborne transmission of diarrhoea-causing enteric pathogens.

## 4. Materials and Methods

### 4.1. Study Context

In Mozambique, the GEMS was conducted by the *Centro de Investigação em Saúde de Manhiça* (CISM), in six health facilities in the Manhiça District [55], which is located approximately 80 km north of the capital city Maputo in the country’s Southern region. The district covers 2380 km^2^ and has a subtropical climate with two distinct seasons: a warm, rainy season from November to April, and the cool and dry season during the rest of the year [17,56]. Since 1996, CISM has been conducting a continuous Health and Demographic Surveillance System (HDSS) with regular update of demographic events for all surveyed population (current population followed: 203,132 uniquely identified individuals; 46,851 enumerated and geo-positioned households; 27,504 are children under 5 years). During the study period, the HDSS was covering approximately 95,000 inhabitants [17].

The rationale, study design, and methodology of the GEMS have been previously described elsewhere [57]; the study comprised three years of recruitment of acute moderate-to-severe diarrhoea cases (MSD, GEMS1) and one additional year including less-severe diarrhoea (LSD, GEMS1A) [4,5]. In Manhiça, the GEMS collected samples uninterruptedly over 5 years, from December 2007 to October 2011, and GEMS1A from November 2011 to November 2012. The standardised epidemiological and clinical methods for the case-control study as well as the full definitions have been previously described elsewhere [18,55]. Briefly, all children aged 0–59 months (stratified in three age groups: 0–11 months, 12–23 months, and 24–59 months), presenting in the six sentinel health facilities with diarrhoea meeting inclusion criteria for the study were invited to participate. Community controls (up to three for MSD cases and one for LSD cases) matched to the index case by age, sex, and neighbourhood were identified using the HDSS databases and enrolled within 14 days after enrolment of the case, and the stool samples were collected and sent to the laboratory at CISM [58].

### 4.2. Stool Collection and Initial Testing

The stool samples collection and processing protocols were also standardised across GEMS sites as described elsewhere [55,58]. Samples were collected in sterile flasks and placed in a refrigerator or in a cool-box with a cooler block (2–8 °C) for up to 6 h until transported to the laboratory. Sample aliquots without preservatives were frozen at −80 °C for further testing. The CRYPTOSPORIDIUM II™ ELISA immunoassay (TECHLAB^®^, Blacksburg, VA, USA) was used as screening method for the specific detection of *Cryptosporidium* spp. following the manufacturer’s instructions.

### 4.3. Molecular Study

#### 4.3.1. DNA Extraction and Purification

Molecular analyses were performed only on stool samples that were *Cryptosporidium*-positive by immunoassay. Genomic DNA was isolated from about 200 mg of faecal material by using the QIAamp DNA Stool Mini Kit (Qiagen, Hilden, Germany) according to the manufacturer’s instructions, except that samples mixed with ASL lysis buffer were incubated for 10 min at 95 °C. Resulting eluates (200 μL in PCR-grade water) were stored at −20 °C and shipped to the Spanish National Centre for Microbiology at Majadahonda (Spain) for downstream molecular analysis.

#### 4.3.2. Molecular Detection and Characterisation of *Cryptosporidium* spp.

As this study was based on *Cryptosporidium*-positive samples by ELISA, to optimise time and resources the following diagnostic and genotyping algorithm was implemented. A nested PCR protocol was initially used to amplify an 870-bp fragment of the *gp60* gene of the parasite as previously described [59]. This approach allowed for the differential diagnosis of *C. hominis* and *C. parvum* (the two *Cryptosporidium* species more prevalent in humans), and for the identification of subtype families within these two species. The outer primers were AL-3531_F (5’-ATAGTCTCCGCTGTATTC-3’) and AL-3535_R (5’-GGAAGGAACGATGTATCT-3’), and the inner primers were AL-3532_F (5’-TCCGCTGTATTCTCAGCC-3’) and AL-3534_R (5’-GCAGAGGAACCAGCATC-3’). Reaction mixtures (50 µL) contained 200 nM of each primer and 2–3 μL of template DNA. Cycling conditions included one step of 94 °C for 5 min, followed by 35 cycles of amplification (denaturation at 94 °C for 45 s, annealing at 59 °C for 45 s, and elongation at 72 °C for 1 min), concluding with a final extension of 72 °C for 10 min. The same conditions were used in the secondary reaction, except that the annealing temperature was 50 °C.

Samples with a negative result by *gp60*-PCR were re-analysed by a nested PCR to amplify a 587-bp fragment of the *ssu* rRNA gene of the parasite [60]. This approach allowed for the detection of low burdens of *Cryptosporidium* infections and for the identification of *Cryptosporidium* species other than *C. hominis* or *C. parvum*. The outer primers were CR-P1 (5’-CAGGGAGGTAGTGACAAGAA-3’) and CR-P2 (5’-TCAGCCTTGCGACCATACTC-3’), and the inner primers were CR-P3 (5’-ATTGGAGGGCAAGTCTGGTG-3’) and CPB-DIAGR (5’-TAAGGTGCTGAAGGAGTAAGG-3’). In all cases, reaction mixtures (50 µL) contained 300 nM of each primer and 3 μL of template DNA. Cycling conditions consisted of one step of 94°C for 5 min, followed by 35 cycles of amplification (denaturation at 94 °C for 40 s, annealing at 50 °C for 40 s, and elongation at 72 °C for 1 min), finalising with a final extension at 72 °C for 10 min.

Samples that were identified by *ssu*-PCR (and Sanger sequencing, see below) as *C. meleagridis* were re-analysed at the *gp60* locus by a nested PCR specifically developed for this *Cryptosporidium* species [21]. This protocol amplifies a 900 bp fragment of the *gp60* gene. The outer primers were CRSout115F (5´-GATGAGATTGTCGCTCGTTATC-3´) and CRSout1328R (5´-AACCTGCGGAACCTGTG-3´), and the inner primers were ATGFmod (5´-GAGATTGTCGCTCGTTATCG-3´) and GATR2 (5´-GATTGCAAAAACGGAAGG-3´). Reaction mixtures (50 µL) contained 250 nM of each primer and 2–3 μL of template DNA. Cycling conditions included one step of 95 °C for 4 min, followed by 35 cycles of amplification (denaturation at 95 °C for 30 s, annealing at 60 °C for 30 s, and elongation at 72 °C for 1 min), concluding with a final extension of 72 °C for 7 min. The same conditions were used in the secondary reaction, except that the annealing temperature was 58 °C.

Nested PCR protocols described above were conducted on a 2720 Thermal Cycler (Applied Biosystems, CA, USA). Reaction mixes always included 2.5 units of MyTAQ^TM^ DNA polymerase (Bioline GmbH, Luckenwalde, Germany), and 5× MyTAQ^TM^ Reaction Buffer containing 5 mM dNTPs and 15 mM MgCl_2_. Laboratory-confirmed positive and negative DNA samples of human origin were routinely used as controls and included in each round of PCR. PCR amplicons were visualised on 2% D5 agarose gels (Conda, Madrid, Spain) stained with Pronasafe nucleic acid staining solution (Conda) and recorded using the MiniBIS Pro system controlled by GelCapture version 7.5.2 software (DNR Bio-Imaging Systems, Jerusalem, Israel). A 100 bp DNA ladder (Boehringer Mannheim GmbH, Baden-Wurttemberg, Germany) was used for the sizing of obtained amplicons. Positive-PCR products were directly sequenced in both directions using the internal primer sets described above. DNA sequencing was conducted by capillary electrophoresis using the BigDye^®^ Terminator chemistry (Applied Biosystems) on an on ABI PRISM 3130 automated DNA sequencer at the Core Genomic Facility of the Spanish National Centre for Microbiology, Majadahonda (Spain). Sequencing reactions were repeated on samples for which genotyping was unsuccessful in the first instance.

The *Cryptosporidium* sequences obtained in this study have been deposited in GenBank under accession numbers MW480826–MW480846 (*gp60* locus) and MW487256–MW487266 (*ssu* rRNA locus).

### 4.4. Data Analysis

#### 4.4.1. Epidemiological Analysis

PCR data were entered in a Microsoft Excel spreadsheet (Redmond, WA, USA) and then checked for accuracy and consistency by independent laboratory personnel. Clinical and demographic data were extracted from the original GEMS dataset. *Cryptosporidium* spp., study groups (diarrhoeal vs. non-diarrhoeal, MSD vs. LSD) age groups, and study years were treated as categorical variables. Differences in frequencies were compared using Chi-squared test or Fisher’s exact test as appropriate. A *p*-value < 0.05 was considered statistically significant. Missing values were excluded from the analyses; thus, denominators for some comparisons may differ. Data analyses were performed in Stata version 14 (StataCorp LP, College Station, TX, USA).

#### 4.4.2. Sequence and Phylogenetic Analysis

Raw sequencing data in both forward and reverse directions were visually inspected using the Chromas Lite version 2.1 sequence analysis program [61]. Special attention was paid to the detection and recording of ambiguous (double peak) positions. The BLAST tool was used to search for identity among sequences deposited in the National Center for Biotechnology Information (NCBI) public repository database [62]. Multiple sequence alignment analyses with appropriate reference sequences were conducted using MEGA 6 to identify *Cryptosporidium* species and to annotate the presence of single nucleotide polymorphisms (SNPs) [63]. *Cryptosporidium hominis* and *C. parvum* subtypes were assigned according to the number of TCA (A), TCG (G), ACATCA/ACATCG (R), and TCTT (T) fragment repeats in the microsatellite region of the *gp60* gene, in accordance with the established nomenclature, as previously described [14].

The evolutionary relationships among the identified *Cryptosporidium* species and subtypes were inferred by a phylogenetic analysis using the neighbor-joining method in MEGA 6 [64]. Only sequences with unambiguous (no double peak) positions were used in the analyses. The evolutionary distances were computed using the Kimura 2-parameter method and modelled with a gamma distribution. The reliability of the phylogenetic analyses at each branch node was estimated by the bootstrap method using 1000 replications. Representative sequences of different *Cryptosporidium* species and subtypes were retrieved from the NCBI database and included in the phylogenetic analysis for reference and comparative purposes.

## 5. Conclusions

This study provides the most comprehensive description of the molecular diversity of the enteric protozoan parasite *Cryptosporidium* spp. in Mozambique to date. Our findings revealed the circulation of at least three *Cryptosporidium* species in young Mozambican children primarily affected with diarrhoea. A high intra-species genetic variability was observed within *C. hominis* (subtype families Ia, Ib, Id, Ie, and If) and *C. parvum* (subtype families IIb, IIc, IIe, and IIi), but not within *C. meleagridis* (subtype family IIIb). No associations between *Cryptosporidium* species/genetic variants and age-related patterns could be demonstrated. The predominance of mainly anthroponotically transmitted *C. hominis* and *C. parvum* IIc strongly suggests that most of the *Cryptosporidium* infections detected in the surveyed paediatric population are of human origin. However, a significant proportion of the infections were caused by host-adapted *Cryptosporidium* species (e.g., *C. meleagridis*) or genetic variants (e.g., *C. parvum* “bovine genotype”) suggesting the occurrence of zoonotic transmission events at an unknown rate. Further molecular epidemiological studies are warranted to assess the actual contribution of livestock, poultry, and other domestic animal species to the environmental (including surface waters intended for human consumption and soils) burden of *Cryptosporidium* oocysts in Mozambique and other African endemic areas.

## Figures and Tables

**Figure 1 pathogens-10-00452-f001:**
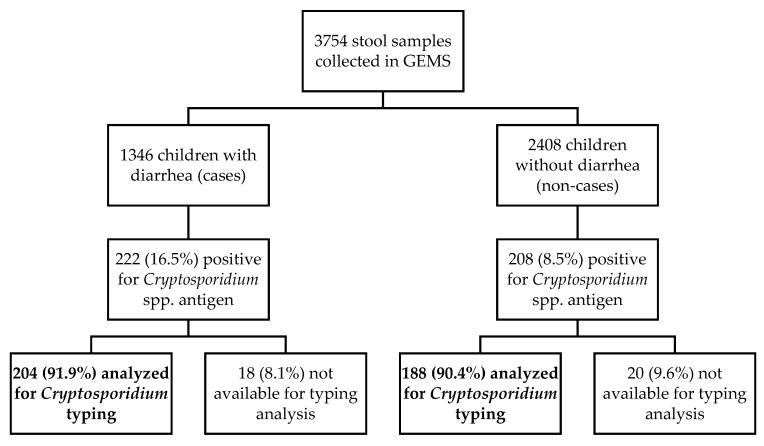
Flow chart summarising the diagnostic and genotyping procedures used in this study.

**Figure 2 pathogens-10-00452-f002:**
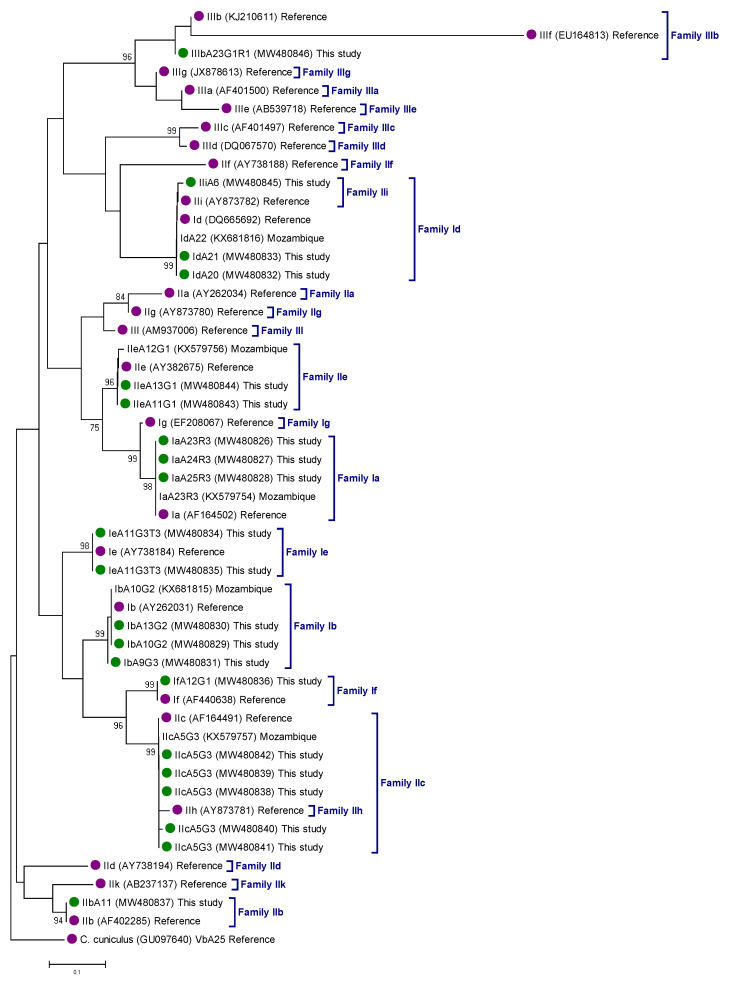
Phylogenetic relationships among *Cryptosporidium hominis* (family I), *C. parvum* (family II), and *C. meleagridis* (family III) genotypes identified in children under 5 years of age recruited during the Global Enteric Multicenter Study at the Manhiça district (Maputo, southern Mozambique), 2007–2012. The analysis was conducted by a neighbor-joining method of the *gp60* gene. Genetic distances were calculated using the Kimura two-parameter model. Green filled dots represent sequences generated in the present study. Purple filled dots represent reference sequences. Bootstrap values lower than 75% are not displayed. *Cryptosporidium cuniculus* was used as outgroup taxon to root the tree.

**Table 1 pathogens-10-00452-t001:** Main epidemiological and clinical variables of *Cryptosporidium*-positive children under five years of age by ELISA (*n* = 392) with diarrhoea (cases) and without diarrhoea (non-cases) according to age group. Children were recruited during the Global Enteric Multicenter Study at the Manhiça district (Maputo, southern Mozambique), 2007–2012.

	0–11 Months		12–23 Months		24–59 Months	
Variable	Cases	Non-Cases ^1^	Cases	Non-Cases ^1^	Cases	Non-Cases ^1^
	*n* = 111 (%)	*n* = 99 (%)	*n* = 72 (%)	*n* = 68 (%)	*n* = 21 (%)	*n* = 21 (%)
MSD ^2^	87 (78.4)	85 (85.9)	45 (62.5)	57 (83.8)	12 (57.1)	19 (90.5)
LSD ^2^	24 (21.6)	14 (14.1)	27 (37.5)	11 (16.2)	9 (42.9)	2 (9.5)
Mean age (months)	7.3	7.1	15.9	16.9	31.1	31
Sex (male)	68 (61.3)	68 (68.7)	48 (66.7)	41 (61.2)	13 (69.9)	14 (66.7)
HIV+ ^3^	10/44 (24.7)	2/18 (11.1)	7/37 (18.9)	2/21 (9.5)	2/9 (22.2)	0 (0.0)
Undernutrition	14 (12.6)	2 (2.0)	12 (16.7)	1 (1.5)	1 (4.8)	0 (0.0)
Co-infections						
Rotavirus	33 (29.7)	13 (13.1)	9 (12.5)	10 (14.7)	4 (19.1)	1 (4.7)
*Shigella* spp.	0 (0.0)	0 (0.0)	6 (8.3)	0 (0.0)	1 (4.8)	0 (0.0)
All ETECs	7 (6.3)	5 (5.1)	13 (18.1)	7 (10.3)	4 (19.1)	4 (19.1)
*G. duodenalis*	16 (14.4)	30 (30.3)	18 (25.0)	38 (55.9)	4 (19.5)	10 (47.6)
*E. histolytica* ^4^	6 (5.5)^4^	8 (8.1)	7 (9.7)	5 (7.4)	2 (9.5)	4 (19.1)

^1^ Non-cases are asymptomatic children without diarrhoea matched by age, sex, and neighbourhood with MSD and LSD cases. ^2^ Only applicable to cases. ^3^ Only part of the participants were tested for HIV, and the numbers of participants with known HIV status are specified in the denominator. ^4^ Missing values: *n* = 1. ETEC: Enterotoxigenic *Escherichia coli*; HIV: Human immunodeficiency virus; LSD: Less severe diarrhoea; MSD: Moderate-to-severe diarrhoea; NA: Not applicable.

**Table 2 pathogens-10-00452-t002:** Diagnostic performance of PCR methods and distribution of the *Cryptosporidium* species detected in children under five years of age (*n* = 396) with diarrhoea (cases) and without diarrhoea (non-cases) according to age group. Children were recruited during the Global Enteric Multicenter Study at the Manhiça district (Maputo, southern Mozambique), 2007–2012.

	0–11 Months			12–23 Months			24–59 Months		
PCR Results	Cases	Non-Cases	*p*	Cases	Non-Cases	*p*	Cases	Non-Cases	*p*
	*n* = 112 (%)	*n* = 101 (%)		*n* = 73 (%)	*n* = 67 (%)		*n* = 21 (%)	*n* = 22 (%)	
*gp60*	49 (44.1)	24 (24.4)	0.003	37 (51.4)	30 (44.1)	0.389	3 (14.3)	3 (14.3)	1
(*n* = 146)									
*ssu rRNA* ^1^	12/62 (19.4)	10/75 (13.3)	0.339	9/35 (25.7)	8/38 (21.1)	0.638	4/18 (22.2)	1/18 (5.6)	0.148
(*n* = 44)									
Both	61 (54.9)	34 (34.3)	0.003	46 (63.9)	38 (55.9)	0.334	7 (33.3)	4 (19.1)	0.292
(*n* = 190)									
Species ^2^									
*C. hominis*	45 (47.4)	27 (29.4)	0.011	33 (55.9)	26 (46.4)	0.308	3 (22.2)	3 (15.0)	0.687
(*n* = 138)									
*C. parvum*	14 (21.9)	6 (8.5)	0.028	12 (31.6)	9 (23.1)	0.402	2 (12.5)	0 (0.0)	0.229
(*n* = 43)									
*C. meleagridis*	1 (1.9)	1 (1.5)	1	1 (3.7)	3 (9.1)	0.620	1 (6.7)	1 (5.6)	1
(*n* = 8)									
Unknown ^3^	1 (1.9)	0 (0.0)	0.440	0 (0.0)	0 (0.0)	NA	0 (0.0)	0 (0.0)	NA
(*n* = 1)									

^1^ Only negative samples by *gp60*-PCR (*n* = 250) were analysed by *ssu*-PCR. ^2^ Species assigned on the combination of both *gp60*-PCR and *ssu*-PCR results. ^3^ Poor sequence quality data only allowed subtyping at genus level. NA: Not applicable.

**Table 3 pathogens-10-00452-t003:** Diversity, frequency, and main molecular features of *Cryptosporidium*-positive samples at the *gp60* and *ssu* rRNA loci in children under 5 years of age recruited during the Global Enteric Multicenter Study at the Manhiça district (Maputo, southern Mozambique), 2007–2012. GenBank accession numbers of representative sequences were provided.

Locus	Species	Isolates	Family	Subtype	Reference	Stretch	Single Nucleotide Polymorphisms	GenBank ID
*gp60*	*C. hominis*	37	Ia	IaA23R3	KX579755	3–805	None	MW480826
		1		IaA24R3	KX579755	1–793	84_86InsTCA	MW480827
		3		IaA25R3	JF927194	18–838	None	MW480828
		7	Ib	IbA10G2	AY262031	22–857	None	MW480829
		15		IbA13G2	MT053132	13–896	G85A	MW480830
		2		IbA9G3	DQ665688	14–825	None	MW480831
		1	Id	IdA20	JX088404	48–904	None	MW480832
		1		IdA21	MN904672	47–910	None	MW480833
		39	Ie	IeA11G3T3	AY738184	19–923	None	MW480834
		1		IeA11G3T3	AY738184	48–923	T284Y, A662R	MW480835
		10	If	IfA12G1	EU161655	1–870	None	MW480836
	*C. parvum*	1	IIb	IIbA11	AY166805	1–782	51_59DelTCATCATCA	MW480837
		11	IIc	IIcA5G3	GU214365	31–851	None	MW480838
		7		IIcA5G3	GU214365	29–854	38 SNPs ^2^	MW480839
		5		IIcA5G3	GU214365	29–853	40 SNPs ^2^	MW480840
		1		IIcA5G3	GU214365	50–853	C110T	MW480841
		1		IIcA5G3	GU214365	50–853	40 SNPs ^2^	MW480842
		1	IIe	IIeA11G1	MN904721	1–813	None	MW480843
		1		IIeA13G1	KU852716	7–795	None	MW480844
		1	IIi	IIiA6-like	AY873782	26–932	85 SNPs ^2^	MW480845
	*C. meleagridis* ^1^	4	IIIb	IIIbA23G1R1	MK331716	1–714	None	MW480846
*ssu* rRNA	*C. hominis*	17	–	–	AF108865	529–954	None	MW487256
		1	–	–	AF108865	587–965	A892R	MW487257
		2	–	–	AF108865	591–969	T795Y, A892R	MW487258
		1	–	–	AF108865	640–956	697delT, T795Y, A892R	MW487259
	*C. parvum*	1	–	–	AF112571	565–956	A646G, T649G, 686_689DelTAAT, A691T	MW487260
		3	–	–	AF112571	526–1039	A646G, T649G, 686_689DelTAAT, T693A	MW487261
		1	–	–	AF112571	524–1039	A646G, 647_649DelATT, T663C, 686_689DelTAAT, C795T	MW487262
		5	–	–	AF112571	539–1031	A646G, T649G, 686_689DelTAAT, T693A, C795T	MW487263
		1	–	–	AF112571	526–965	A646G, T649G, 686_689DelTAAT, T693A, C795Y	MW487264
		3	–	–	AF112571	539–954	A646G, T649G, 686_689DelTAAT, A691T, C795Y, A892R	MW487265
	Unknown	1	–	–	–	–	–	–
	*C. meleagridis*	8	–	–	AF112574	524–1034	None	MW487266

^1^ Samples initially diagnosed by *ssu*-PCR and subsequently genotyped at the *gp60* locus using the *C. meleagridis*-specific PCR protocol described elsewhere [21]. ^2^ See details in Appendix A. Del: nucleotide(s) deletion(s); NA: Not applicable; Y: C/T; R: A/G. Novel genotypes are shown underlined.

**Table 4 pathogens-10-00452-t004:** Diversity and frequency of *Cryptosporidium* subtypes families within *C. hominis*, *C. parvum*, and *C. meleagridis* in symptomatic (cases) children under 5 years of age according to severity of the diarrhoea, age group, and HIV coinfection. Children were recruited during the Global Enteric Multicenter Study at the Manhiça district (Maputo, southern Mozambique), 2007–2012.

		*C. hominis*					*C. parvum*	*C. meleagridis*
Variable	Total	Ia	Ib	Id	Ie	If	IIc	IIIb
	*n* = 90 (%)	*n* = 25 (%)	*n* = 11 (%)	*n* = 2 (%)	*n* = 28 (%)	*n* = 6 (%)	*n* = 17 (%)	*n* = 1 (%)
Diarrhoea								
MSD	60 (66.7)	11 (44.0)	7 (63.6)	1 (50.0)	26 (92.9)	5 (83.3)	9 (52.9)	1 (100)
LSD	30 (33.3)	14 (56.0)	4 (36.4)	1 (50.0)	2 (7.1)	1 (16.7)	8 (47.1)	0 (0)
Age (months)								
0–11	49 (54.4)	16 (64.0)	8 (72.7)	1 (50.0)	14 (50)	2 (33.3)	8 (47.1)	0 (0)
12–23	37 (41.1)	7 (28.0)	3 (27.3)	1 (50.0)	13 (46.4)	4 (66.7)	9 (52.9)	0 (0)
24–59	4 (4.4)	2 (8.0)	0 (0.0)	0 (0.0)	1 (3.6)	0 (0.0)	0 (0.0)	1 (100)
Co-infections								
HIV+ ^1^	9/42 (21.4)	5/17 (29.4)	1/4 (25)	0/1 (0.0)	1/5 (20)	0/1 (0.0)	2/14 (14.3)	NA

^1^ Frequencies calculated over the total of HIV+ children only. 42 of the 90 children had an HIV test result. HIV: Human immunodeficiency virus; LSD: Less severe diarrhoea; MSD: Moderate-to-severe diarrhoea. NA: Not applicable.

## Data Availability

All relevant data are within the paper and its Appendix A.

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
