# Peer review of "Molecular Characterisation of Cryptosporidium spp. in Mozambican Children Younger than 5 Years Enrolled in a Matched Case-Control Study on the Aetiology of Diarrhoeal Disease"

_pathogens, 2021, doi:10.3390/pathogens10040452_

Round 1

Reviewer 1 Report

line 88: the IIc family is predominant in Africa but it depends of geography; in Europe the most dominant C. parvum subtype is strongly the IIa (see Cacciò SM, Chalmers RM. Human cryptosporidiosis in Europe. Clin Microbiol Infect. 2016 Jun;22(6):471-80. doi: 10.1016/j.cmi.2016.04.021. Epub 2016 May 10. PMID: 27172805.)

Fihure 1 : please described why specimen were not available for typing ?

Regarding rare species detection, ELISA methods are known to be poorly sensitive for detection of rare Cryptosporidium species, it should be discussed to explicit why only C. meleagridis was identified in term of rare species

Author Response

Referee #1

  1. Line 88: the IIc family is predominant in Africa but it depends of geography; in Europe the most dominant parvum subtype is strongly the IIa (see Cacciò SM, Chalmers RM. Human cryptosporidiosis in Europe. Clin Microbiol Infect. 2016 Jun;22(6):471-80. doi: 10.1016/j.cmi.2016.04.021. Epub 2016 May 10. PMID: 27172805.)

Reply: We thank referee #1 for his/her comment. Please note that we have already acknowledged this point in current lines 302-304 of the discussion section, where we purposely stated that “Taken together, these geographically segregated patterns of C. parvum genetic variants may be indicative of differences in sources of infections and transmission pathways”. This very same point has been also highlighted by Squire and Ryan in their thorough revision on the molecular epidemiology of Cryptosporidium infections in humans, animals and environmental sources in African countries (see current reference #11 Squire, S.A.; Ryan, U. Cryptosporidium and Giardia in Africa: current and future challenges. Parasit Vectors 2017, 10, 195). This reference has been mentioned and discussed several times in the present manuscript.

  1. Figure 1 : please described why specimen were not available for typing ?

Reply: We thank referee #1 for his/her comment. As mentioned in the manuscript, in this study we retrospectively analysed samples collected between 2007 and 2012 in the context of the Global Enteric Multicenter Study (GEMS). Along this period, those samples have been used for analytical purposes several times, including the initial identification of diarrhoea-causing agents reported in previous studies (Kotloff et al. Lancet 2013;382:209-222, Nhampossa et al. PLoS One 2015;10:e0119824., and Kotloff et al., Lancet Glob Health 2019;7:e568-e584, among others). As a result, some samples have been completely depleted and were not available for the present study. This point has been now acknowledged at the end of the first paragraph in subsection 2.1 of the Results section.

  1. Regarding rare species detection, ELISA methods are known to be poorly sensitive for detection of rare Cryptosporidium species, it should be discussed to explicit why only meleagridis was identified in term of rare species

Reply: We thank referee #1 for his/her comment. We have acknowledged this possibility as a potential limitation of the study in the paragraph devoted to this issue in the Discussion section.

Reviewer 2 Report

The manuscript described study conducted on cryptosporidiosis among Mozambian children younger than 5 years. Futhermore, genetic diversity and frequency of Cryptosporidium species and subtypes were assessed.

In my opinion the article provide important knowledge in the field of epidemiology. Futhermore, the authors suggested that most of the Cryptosporidium infections were anthroponotically transmitted which is very interesting in the future study.

However, I have 2 minor things for the authors:

  1. Please, add „95% confidence interval” in tab.1 and tab.2.
  2. The full parasites names should be italic, especially in the part of Results (line 137-154)

Apart from 2 issues I have no suggestions for improvement of this paper.

Author Response

Referee #2

The manuscript described study conducted on cryptosporidiosis among Mozambican children younger than 5 years. Furthermore, genetic diversity and frequency of Cryptosporidium species and subtypes were assessed. In my opinion the article provide important knowledge in the field of epidemiology. Furthermore, the authors suggested that most of the Cryptosporidium infections were anthroponotically transmitted which is very interesting in the future study.

We thank Referee #2 for his/her favourable preliminary comments on our manuscript.

However, I have 2 minor things for the authors:

  1. Please, add „95% confidence interval” in tab.1 and tab.2.

Reply: please note that the above-mentioned tables summarize the main epidemiological and clinical variables for Cryptosporidium-positive children (Table 1) and the distribution of the Cryptosporidium species found according to the age of the participants (Table 2). These are descriptive tables were no prevalence data are shown, and no statistical analyses were conducted or odds ratios estimated. Therefore, there is no need of presenting 95% confidence intervals in these tables.

  1. The full parasites names should be italic, especially in the part of Results (line 137-154)

Reply: We thank referee #2 for this observation. All the appropriate modifications have been done on the manuscript, where this and other changes have been highlighted in red for better identification.

Reviewer 3 Report

The manuscript entitled “Molecular characterization of Cryptosporidium spp. in Mozambican children younger than 5 years enrolled in a matched case control study on the aetiology of diarrhoeal disease” is an impressive, excellent survey on the genetic structure and epidemiology of Cryptosporidium spp. in children in Mozambique. It is very well presented in the manuscript and of course, I find it an outstanding work deserving publication.

I only have the following minor comments for the authors to consider:

Abstract:  For the sentence “Children <23 months were more exposed to Cryptosporidium spp. infections than older children” to be justified here, the initial part of the survey, i.e. the screening of 3,754 stool samples needs to be mentioned in the abstract.

Lines 48-49: “9% of the 5.8 million deaths” is this referring to children mortality as the begging of the sentence? If not rephrase to clarify.

Line 258: “including” to be changed with “i.e.” or “that is”

Lines 270-271: “However, we cannot rule out the possibility that these infections were acquired indirectly through the consumption of contaminated water or food”. To my opinion this sentence should not begin with “however”, as an infection by a parasite of animal origin is still considered of zoonotic nature when occurs via contaminated water or food (which is a major way of human infection anyways). So, the sentence does not contain any contradiction to justify a “however” introduction. Nevertheless, human-to-human transmission of ”animal” genotypes can also occur, and it is questionable if these infections could be considered of zoonotic nature.

Line 326: Do the authors mean false-negative by PCR? Please clarify. If this is the case, it is not clear why would these species be missed by ssu rRNA PCR.

Author Response

Referee #3

The manuscript entitled “Molecular characterization of Cryptosporidium spp. in Mozambican children younger than 5 years enrolled in a matched case control study on the aetiology of diarrhoeal disease” is an impressive, excellent survey on the genetic structure and epidemiology of Cryptosporidium spp. in children in Mozambique. It is very well presented in the manuscript and of course, I find it an outstanding work deserving publication.

We thank Referee #3 for his/her favourable preliminary comments on our manuscript.

I only have the following minor comments for the authors to consider:

  1. Abstract: For the sentence “Children <23 months were more exposed to Cryptosporidium infections than older children” to be justified here, the initial part of the survey, i.e. the screening of 3,754 stool samples needs to be mentioned in the abstract.

Reply: Following referee #3 recommendation, this point has been now mentioned in line 29 of the Abstract section.

  1. Lines 48-49: “9% of the 5.8 million deaths” is this referring to children mortality as the begging of the sentence? If not rephrase to clarify.

Reply: Yes, the figures indicate children mortality rates caused by diarrhoeal diseases. To improve clarity, the sentence has been now rephrased as “…for approximately 9% of the 5.8 million deaths associated to this condition reported in 2015”.

  1. Line 258: “including” to be changed with “i.e.” or “that is”

Reply: to avoid misleading interpretations, this part of the sentence has been now expressed between brackets in lines 257-258.

  1. Lines 270-271: “However, we cannot rule out the possibility that these infections were acquired indirectly through the consumption of contaminated water or food”. To my opinion this sentence should not begin with “however”, as an infection by a parasite of animal origin is still considered of zoonotic nature when occurs via contaminated water or food (which is a major way of human infection anyways). So, the sentence does not contain any contradiction to justify a “however” introduction. Nevertheless, human-to-human transmission of ”animal” genotypes can also occur, and it is questionable if these infections could be considered of zoonotic nature.

Reply: the referee raised an important issue. To avoid misleading interpretations, the whole paragraph has been now rewritten as “This is without precluding that some of the infections caused by this genetic variant of C. parvum may be also transmitted through person-to-person contact. The extent of the exact contribution of each potential transmission pathway (zoonotic, anthropic, direct contact, or indirect through ingestion of contaminated water or food) remains to be elucidated”.

  1. Line 326: Do the authors mean false-negative by PCR? Please clarify. If this is the case, it is not clear why would these species be missed by ssu rRNA PCR.

Reply: please see our answer to comment #3 raised by Referee #1 in current lines 327-330 of the Discussion section.